# Green Turtle Fibropapillomatosis: Tumor Morphology and Growth Rate in a Rehabilitation Setting

**DOI:** 10.3390/vetsci10070421

**Published:** 2023-06-29

**Authors:** Costanza Manes, Richard M. Herren, Annie Page, Faith D. Dunlap, Christopher A. Skibicki, Devon R. Rollinson Ramia, Jessica A. Farrell, Ilaria Capua, Raymond R. Carthy, David J. Duffy

**Affiliations:** 1One Health Center of Excellence, University of Florida, Gainesville, FL 32611, USA; c.manes@ufl.edu; 2Department of Wildlife Ecology and Conservation, University of Florida, Gainesville, FL 32611, USA; rherren@ufl.edu (R.M.H.); ngosi@ufl.edu (R.R.C.); 3The Sea Turtle Conservancy, Gainesville, FL 32609, USA; 4Harbor Branch Oceanographic Institute, Florida Atlantic University, Fort Pierce, FL 34946, USA; cpagekarjian@fau.edu; 5Department of Biology, University of Florida, Gainesville, FL 32611, USA; dunlap.faith@ufl.edu (F.D.D.); duffy@whitney.ufl.edu (D.J.D.); 6Fisheries and Aquatic Sciences, University of Florida, Gainesville, FL 32611, USA; c.skibicki@ufl.edu; 7The Whitney Laboratory for Marine Bioscience and Sea Turtle Hospital, University of Florida, St. Augustine, FL 32080, USA; devonrenee@whitney.ufl.edu (D.R.R.R.); jessicafarrell@ufl.edu (J.A.F.); 8School of Advanced International Studies, John Hopkins University, 40126 Bologna, Italy; 9Institute of Food and Agricultural Sciences, University of Florida, Gainesville, FL 32611, USA; 10U.S. Geological Survey, Florida Cooperative Fish and Wildlife Research Unit, University of Florida, Gainesville, FL 32611, USA

**Keywords:** green turtles, fibropapillomatosis, smooth tumors, rugose tumors, growth rate, progression, regression, rehabilitation

## Abstract

**Simple Summary:**

Green turtles (*Chelonia mydas*) are globally afflicted by a tumoral disease called fibropapillomatosis (FP). Affected turtles experience growth of tumors on various parts of their body, including skin tissue on the flippers and neck as well as in the eyes and mouth. Internal tumors also occur sometimes, for example, on heart, lungs, and kidneys. Since FP was first described in 1938, FP tumors have been categorized into two main morphological types: rugose and smooth tumors. Rugose tumors are characterized by papillose structures and a rough texture, while smooth tumors have a smoother and more even surface and texture. It has often been suggested in the literature that, while rugose tumors tend to show active growth, smooth tumors might be a sign of disease regression. However, this hypothesis warrants further verification. In our study, we tracked and compared the growth of rugose and smooth tumors across nine green turtles in rehabilitation at the University of Florida Whitney Laboratory Sea Turtle Hospital in St. Augustine, Florida. According to our findings, rugose FP tumors grow at significantly faster rates than smooth ones, but both FP tumor morphologies still show a general progression pattern. Our study is, to our knowledge, the first-ever assessment of growth-rate differences between smooth and rugose FP tumors and offers important preliminary data to answer a long-standing question in FP research.

**Abstract:**

Fibropapillomatosis (FP) is a neoplastic disease most often found in green turtles (*Chelonia mydas*). Afflicted turtles are burdened with potentially debilitating tumors concentrated externally on the soft tissues, plastron, and eyes and internally on the lungs, kidneys, and the heart. Clinical signs occur at various levels, ranging from mild disease to severe debilitation. Tumors can both progress and regress in affected turtles, with outcomes ranging from death due to the disease to complete regression. Since its official description in the scientific literature in 1938, tumor growth rates have been rarely documented. In addition, FP tumors come in two very different morphologies; yet, to our knowledge, there have been no quantified differences in growth rates between tumor types. FP tumors are often rugose in texture, with a polypoid to papillomatous morphology, and may or may not be pedunculated. In other cases, tumors are smooth, with a skin-like surface texture and little to no papillose structures. In our study, we assessed growth-rate differences between rugose and smooth tumor morphologies in a rehabilitation setting. We measured average biweekly tumor growth over time in green turtles undergoing rehabilitation at the University of Florida Whitney Laboratory Sea Turtle Hospital in St. Augustine, Florida, and compared growth between rugose and smooth tumors. Our results demonstrate that both rugose and smooth tumors follow a similar active growth progression pattern, but rugose tumors grew at significantly faster rates (*p* = 0.013) than smooth ones. We also documented regression across several examined tumors, ranging from −0.19% up to −10.8% average biweekly negative growth. Our study offers a first-ever assessment of differential growth between tumor morphologies and an additional diagnostic feature that may lead to a more comprehensive understanding and treatment of the disease. We support the importance of tumor morphological categorization (rugose versus smooth) being documented in future FP hospital- and field-based health assessments.

## 1. Introduction

Fibropapillomatosis (FP) is a neoplastic disease affecting all sea turtles species; yet, it is most commonly found in green turtles (*Chelonia mydas*) [1]. Since its first official report in 1938, the disease is now spread globally and mostly affects juvenile green turtles [2,3]. A positive association was reported between FP and chelonid herpesvirus 5 (ChHV5), but no recent viral mutation was found to explain recent disease incidence [4]. Clinical signs of FP can often be debilitating. The disease can result in cutaneous tumors arising from soft tissues, and these can occur anywhere on the body, including the flippers, inguinal area, mouth, and eyes. Tumoral lesions can also grow internally on organs such as the lungs, kidney, and heart, which may eventually kill the afflicted individual [5]. Phylogenetic data indicate that green turtles acquire FP upon recruitment to neritic waters [6]. Tumors are infrequently observed in adult turtles, possibly because either severely affected individuals died as juveniles, tumors regressed after a successful immune response, or a combination of these factors [7,8]. Ongoing research is attempting to answer the question of the disease’s true consequences for the population. In a cohort of 61 recaptured FP-affected green turtles in Brazil, 72.1% showed tumor progression, 32.8% showed tumor regression, and 24.6% showed both progression and regression [9]. Differences between progression and regression patterns were previously found among different age cohorts of green turtles in Hawaii, with adult nesters reporting a significantly higher rate of FP tumor regression compared to juveniles [10]. There are further reports of spontaneous tumor regression [11], and some modeling studies suggest that most affected individuals eventually recover from FP [12]. However, this recovery assessment was based on 72 animals captured with FP and subsequently re-captured without FP, and there was no firm input in the modelling of how many turtles with FP subsequently died and therefore were not recaptured, as this is an unknown variable in the dataset [12]. Therefore, there is generally a lack of published data on turtles that had FP but were not monitored or re-captured in the wild, potentially skewing any modelling or observation towards only those individuals that survived to be re-captured. FP-related mortality certainly occurs both in hospitals and in the wild [13,14,15,16,17]. One recent study found that death occurred in more than 75% of FP-afflicted green turtles admitted into rehabilitation facilities [18]. Health assessments in many bays and lagoons in Florida have consistently reported FP prevalence at or above 50% in the last decades [7,19,20,21,22,23]. This high occurrence is problematic, as it could indicate that a small change, such as an outbreak of a different disease, shifting forage availability, or other environmental perturbance (e.g., harmful algal bloom, major storm, or other climatic event), could have important, population-wide consequences on the viability of green turtles with FP.

Since FP was first characterized, lesions have been described in a wide variety of sizes, shapes, and morphologies. FP tumor morphology is typically described as either smooth or rugose (Figure 1) [24] and is likely related to the cellular composition of the tissues from which the tumors arise [25,26]. In FP’s first description from 1938, researchers outlined a clear difference from the “outstanding” (i.e., rugose) tumors and the “smooth oval or round” (i.e., smooth) tumors [2]. Recently, research has provided more thorough descriptions of FP tumor morphology, such as “prominent connective tissue matrix with proliferating fibroblasts” [27], “small, round, raised, white areas” [28], “pigmented brown to dark grey or black with rough and papillary surfaces”, and “appearance with a fibromatous surface” [29]. Gross examination has been the primary way to describe and differentiate tumors based on their morphology, although methods for this are not standardized, and morphological categorization of observed tumors is not routinely or consistently performed in FP-severity assessments [30,31,32]. Previous green turtle mark–recapture studies have postulated that tumor texture is related to stage(s) of tumor development such that active/developing tumors have a rugose, papillomatous texture, while regressed tumors tend to exhibit a smooth surface [20]. One reason for this inference is the observation that rugose FP tumors that might indicate a progression of the disease are often observed in juvenile green turtles, while smooth FP tumors with a smooth appearance that might characterize regression are often observed in sub-adults and adults (Herren, unpublished data). However, this hypothesis that progression and regression relate to tumor morphology has not been further tested or verified in a controlled setting.

Generally, FP tumor growth-rate patterns in hospitalized green turtles might have a role in disease severity and outcome [33]. To our knowledge, there have been no clinical studies on FP tumor growth rates with active differentiation of lesions based on morphological characteristics. In this study, we selected green turtles with FP in rehabilitation at the University of Florida Whitney Laboratory Sea Turtle Hospital (UFWLSTH) exhibiting tumors of both rugose and smooth morphology. We hypothesize that FP tumors belonging to different morphology might show different growth patterns in the affected animals. To test this in the current study, we aimed to track the growth of the different FP tumor morphologies (rugose versus smooth) over time with image scanning software ImageJ (https://imagej.nih.gov/ij, accessed on 10 July 2022).

## 2. Materials and Methods

### 2.1. Study Individuals Selection

Our study is based on previously described methodology that used ImageJ to track FP tumor growth rates in hospitalized individuals [33]. We calculated FP tumor growth based on measurements from photographs of nine rehabilitating green turtles (*Cm* 1–9) hospitalized at the UFWLSTH during 2015–2021. Selection criteria included green turtles with both rugose and smooth external FP tumors. Photographs were taken using an Olympus Tough TG-4 held approximately 30 cm away from each tumor and with a scale in frame. At the time of admission, 4/9 turtles (44%) were emaciated, 1/9 (11%) was in good body condition, and 4/9 (44%) were in robust condition. Mean (±standard deviation) curved carapace length was 39.1 ± 11.9 cm, and mean body mass was 7.8 ± 9.8 kg. Prior to transfer to rehabilitation, all nine turtles were found stranded along the east coast of Florida, between Anastasia State Park (north 29.870729, west −80.274881) and Sykes Creek Bridge (north 28.361066, west −80.678852). Four out of nine (44%) turtles died or were euthanized, and five (56%) were released after successful rehabilitation.

### 2.2. Tumor Categorization by Morphology and Tumor Area Measurement

Gross images of turtles were visually scanned for the presence of external tumors. Using ImageJ software, each tumor was selected for size measurement and labeled with a unique ID number, anatomic location, date of photograph, and tumor morphology (rugose or smooth). If tumors were surgically removed during the turtle’s hospital stay, “NA” was added in their tumor measurement data, and further growth values were excluded from the analysis after the date of tumor removal. Morphology characterization was assessed by gross examination. Verrucous and/or papillomatous tumors were labeled as “R” (rugose), and tumors with a smooth texture were labeled as “S” (smooth). All nine turtles had at least one rugose tumor and at least one smooth tumor. A scale was set for accurate pixel estimate in ImageJ using the “Set Scale” function on the 25 cm scale bar present in each photograph. Once each photograph was scaled, tumors were individually measured in centimeters and their area recorded in chronological order.

### 2.3. Average Biweekly Growth Calculation and Statistical Analyses

Growth was calculated for each tumor via the *dyplr* function using the statistical software R (R Core Team 2021. R: A language and environment for statistical computing. R Foundation for Statistical Computing, Vienna, Austria. Retrieved from https://www.R-project.org/, accessed on 15 January 2022). For 7/9 turtles, photographs were taken once every two weeks, from intake until their release or death. We use the term “biweekly” throughout this paper to indicate the temporal interval of “every two weeks”. Hence, ImageJ photograph-based measurements were used to extract average biweekly growth for 7/9 turtles. For 2/9 turtles (*Cm* 2 and *Cm* 7), intervals between photographs were longer (60–100 days); hence, overall change in tumor size (%) was calculated and then averaged biweekly for data consistency. Each turtle’s tumor areas in centimeters were plotted in chronological order to observe size increase (progression) and/or size decrease (regression) and differentiated based on morphology (rugose and smooth) (Figure 2A–I). Because the data were non-normally distributed, and the sample sizes were small, Kruskal–Wallis tests were used to compare changes in tumor area between rugose and smooth tumors using the statistical software R. Chi-square tests were used to evaluate whether tumor morphology might be related to the anatomic region of tumor growth, using the statistical software R.

## 3. Results

### 3.1. FP Tumor Progression and Regression Patterns per Turtle

For each turtle, total FP tumors were counted, categorized based on morphology (rugose or smooth), and their growth tracked (Figure 2A–I). Mean percentage change in size +/− standard deviation were calculated for each examined tumor categorized by individual and tumor morphology (rugose and smooth) (Table 1). All turtles showed at least one rugose tumor and one smooth tumor, with total FP tumors per turtle ranging from *n* = 2 (*Cm* 5, 7, and 9) up to *n* = 18 (*Cm* 4). Overall, there were more rugose (*n* = 40) than smooth (*n* = 17) tumors across the examined turtles. Of all tumors analyzed in our study (*n* = 57), 17.5% showed regression patterns, and 82.4% showed progression patterns. Across all tumors, regression ranged from a −10.8% to −0.19% average biweekly change in size, while progression ranged from a 0.32% to 108.2% average biweekly change in size. Across rugose tumors, biweekly average change in size ranged from −10.8% to 71.2%. Across smooth tumors, biweekly average change in size ranged from −10.1% to 108.2%. All tumors analyzed were anatomically distributed among four main anatomic regions: 5/57 (9%) along (front or hind) flippers, 40/57 (70%) on the base of (front or hind) flippers, 11/57 (19%) on plastron, and 1/57 (2%) on ventral neck (Table 2).

### 3.2. Statistical Analysis Output

A Shapiro–Wilk test was used to evaluate data normality (Shapiro–Wilk test, W = 0.88, *p* < 0.01). A Kruskal–Wallis test on non-normally distributed data revealed that rugose FP tumors had significantly higher growth than smooth tumors (χ^2^ = 6.13, df = 1, *p* = 0.013) (Figure 3). A chi-square test revealed no significant correlation between FP tumor morphology and anatomic region of tumors (χ^2^ = 5.502, df = 3, *p* > 0.05).

## 4. Discussion

Fibropapillomas have been reported in other animal species, including rabbits, horses, dogs, sheep, deer, mice, and birds [34], with characterization of rugose and smooth fibropapillomas in cattle [35]. To our knowledge, this is the first study to report differences in growth between the two most common FP tumor morphologies observed in afflicted green turtles. We found a significant difference (*p* = 0.013) in biweekly growth between FP tumors morphologies among nine rehabilitating green turtles, with rugose tumors growing significantly faster than smooth tumors. Overall, five turtles had more rugose FP tumors than smooth ones, three had an equal number of smooth and rugose tumors, and one had twice the number of smooth tumors as rugose ones. The turtles analyzed in this study showed high individual variability in tumor growth rates. When comparing percent change in tumor size based on tumor morphology, standard deviation from the mean was slightly higher between smooth tumors (*SD* = 32.2) than between rugose tumors (*SD* = 17.6), reflecting greater variation in smooth lesions between individual turtles. This is reflected in the unusually high biweekly change in tumor size (>75%) having originated mostly from smooth FP tumors, indicating that those lesions may, in some cases, show more extreme measurements of progression (i.e., 108.2% and 76.5%) than rugose FP tumors. The highest recorded change in tumor size measurement (108.2%) belonged to *Cm* 9, which reported a total of two FP tumors, one rugose and one smooth. Despite the high progression in the smooth tumor, progression in *Cm* 9′s rugose tumor was much lower (13.9%), which may highlight a relevant growth pattern difference between FP tumor morphologies within a single individual. 

In terms of mortality, five turtles were released a few months to a year following their intake date, and four died or were euthanized. Among the four deceased turtles, *Cm* 3 and 4 had the highest number of rugose tumors (*n* = 14), and *Cm* 4 also had the second-highest number of smooth tumors (*n* = 4) and the highest overall number of tumors (*n* = 18). The other two deceased turtles, *Cm* 5 and 6, were found to have FP masses in their lungs upon necropsy. Most (4/6) FP tumors that grew more than half their size (>50%) were rugose, which might explain our significant findings of rugose tumors growing significantly faster than smooth ones on an average biweekly basis. The two FP tumors that showed the highest biweekly regression (>−10%) in our study were one rugose tumor at −10.8% (*Cm* 2) and one smooth tumor at −10.1% (*Cm* 8). At the time of intake (first photograph), *Cm* 2’s regressing rugose tumor and *Cm* 8’s regressing smooth tumor were extremely similar in size (1.75 cm and 1.85 cm, respectively). Aware of the possible influence of individual variability between *Cm* 2 and 8, we report that in this case, a smooth FP tumor did not regress at higher rates compared to a rugose one of similar starting size and under the same rehabilitation settings. 

Despite these observations, positive biweekly growth comprised both rugose and smooth lesions, with rugose tumors generally growing at higher rates than the smooth ones, but with large inter-tumor variation within each category. Researchers who have captured green turtles in the wild have suggested that smooth FP lesions were regressing, while rugose tumors were interpreted to be actively growing [20]. Here, we provide the first quantitative evidence that morphology is not a strict determinant of FP regression/progression patterns, with smooth tumors observed both regressing and progressing. Overall, however, smooth tumors grew significantly less than rugose ones. We also highlight that both growing and regressing tumors can be simultaneously present on the same individual at the same time; therefore, not only do some smooth tumors grow, but the presence of some smooth tumors cannot be taken as a proxy for regression of all tumors on a single individual. Further research also highlights what has been reported for FP tumors overall [33,36], although only anecdotally for rugose versus smooth tumors, that inter-tumor growth variability can be rather high even between tumors on the same individual. The observations presented here highlight the need for further study of morphological differences in FP tumors, including evaluating differences in ChHV5 viral load between smooth and rugose lesions collected from a single turtle. Future investigation on the viral and cellular mechanics potentially underlying the different growth rates between rugose and smooth FP tumors found here could offer a more comprehensive understanding of disease development from the time of diagnosis.

Tumor morphology and anatomic location of tumor (Table 2) were not significantly correlated. A previous study that focused on internal tumors found a significant correlation between tumor gross morphology (firm versus cystic) and the organ affected [5]. To our knowledge, there is no research reporting associations between external FP tumor morphology (rugose and smooth) and anatomic region affected. However, FP tumor morphology could be a factor in tumor size and/or timing of tumor development [37,38]. Among a cohort of FP-afflicted green turtles sampled in Hawaii in 1991, smaller tumors were characterized by a rough surface and dark coloration, while larger tumors were characterized as either rugose or smooth [38]. Another study reported a rough and dark-colored surface of small, presumably early tumors in FP-afflicted green turtles from Florida. Over six months of monitoring captive turtles with FP in that study, tumors were observed to initially develop with a rugose texture, then gradually became less rugose and ulcerated over time [37]. Here, we did not observe any temporal changes in tumor morphology even in turtles hospitalized for several months, indicating that, at least in our sample pool, temporal development was not a predictor of tumor morphological characteristics. This observation could have been influenced by surgical laser interventions on FP tumors, which might have removed the external tumors under study before relevant morphological changes, if any, could be detected.

A larger sample size could provide a more reliable analysis of rugose and smooth tumor growth in future studies on FP tumor morphology. Anecdotally, smooth tumors are denser than rugose tumors, which resemble more folded tissue when dissected. This difference warrants further investigation, as density may affect tumor surface growth rates. Moreover, the results presented here are to be considered within the context of the rehabilitation setting. A turtle’s overall health status is an important factor to consider when interpreting findings from our study, particularly because this work was conducted solely on rehabilitating turtles, therefore providing a conservative yet useful baseline that could benefit from further research. While our study provides evidence that rugose FP tumors grow generally faster than smooth ones, further studies on free-ranging turtles collected through capture–mark–recapture methodology would help expand our understanding of FP progression and regression, including the potential influence of anatomic region, tumor growth rate, and tumor morphology in wild, FP-afflicted sea turtles.

## 5. Conclusions

Current FP diagnostics and severity score methodology present in the literature do not take into account tumor morphology when mapping disease severity [31,32]. We deem this factor rather important to improve our understanding of FP burden inclusive of potential influence of anatomic placement, growth rate, and body conditions in relation to the ratio of smooth or rugose tumors present in an individual. Recording this type of variable (rugose versus smooth) can be highly beneficial to FP research, as the different behavior reflected by the dual morphology could affect the observed severity. Instead of just reporting FP severity score, future research may be able to document progression and regression rates based on morphology alone, which would help us understand disease severity at the time of turtle admission and/or capture. Our investigation, the first to our knowledge, differentiating growth between rugose and smooth tumors can set the baseline for a research series on an important and underestimated factor in FP pathogenesis. Inclusion of a tumor morphological category in FP reports will elucidate an important factor within a well-studied yet still mysterious sea turtle epizootic.

## Figures and Tables

**Figure 1 vetsci-10-00421-f001:**
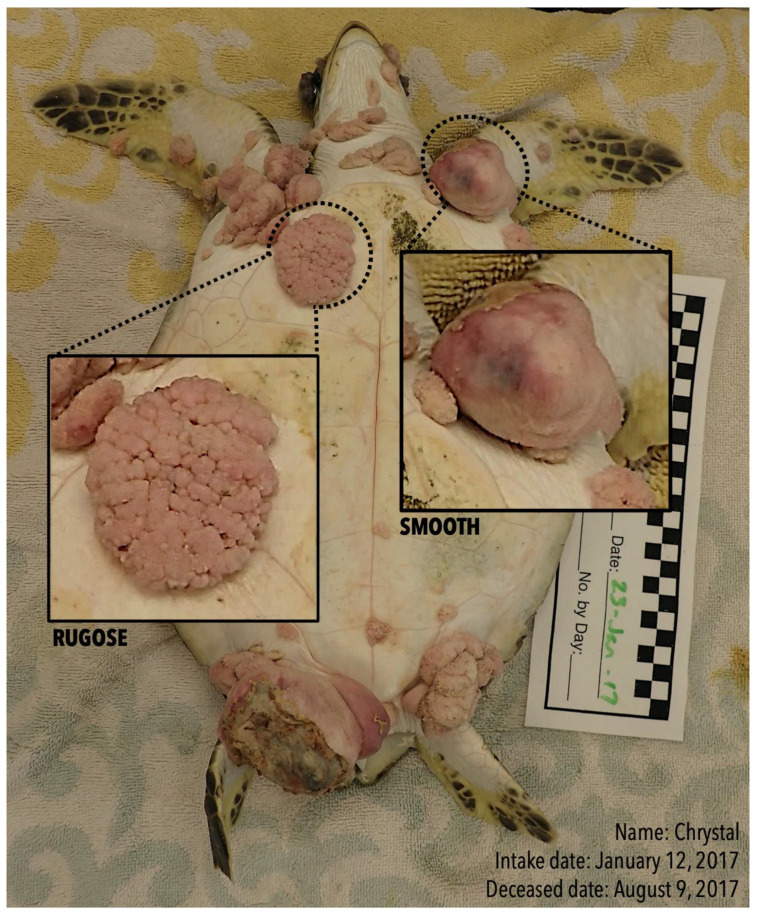
Example of rugose and smooth tumors on a study individual. Photograph of *Cm* 4 (named Chrystal) taken on 23 January 2017. Close-up squares are provided in the image to highlight the difference between two tumors belonging to different morphologies and similar in size. The rugose tumor (left side) presents papillose structure and a rough surface, while the smooth tumor (right side) has a skin-like texture with no papillose structures and appears to present partial necrosis (yellow surface near neck). *Cm* 4 was admitted at the UFWLSTH on 12 January 2017 and deceased on 9 August 2017. Several internal tumors were found upon performing necropsy.

**Figure 2 vetsci-10-00421-f002:**
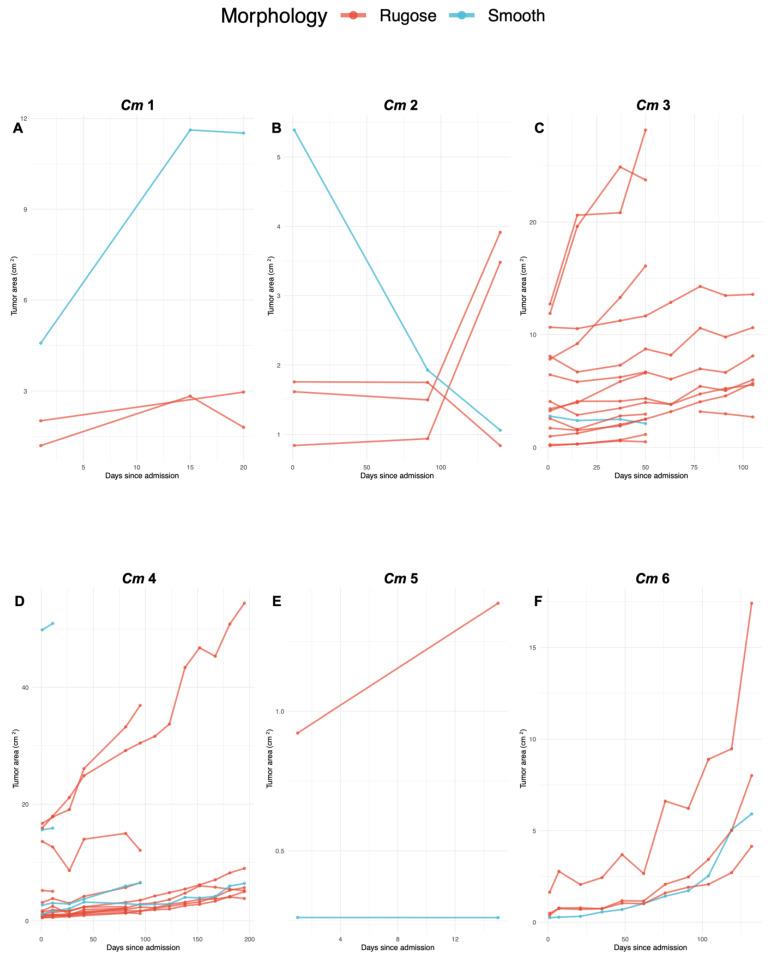
FP tumor growth rate by morphology for each green turtle (*Chelonia mydas*) included in our study. Plots A to I (respectively, *Cm* 1 to 9) show growth rate outcomes of each examined turtle in numerical order. Tumor area in square centimeters is shown on the *y*-axis, and time (days since admission) is shown on the *x*-axis. Axes are scaled to each individual separately. The legend above indicates tumor morphology for rugose (red lines) and smooth (blue lines) FP tumors. Tumor growth was calculated on a biweekly average basis. Lines trending upwards represent tumor progression, while lines trending downward represent tumor regression. Surgically removed tumors were labeled as “NA” and were no longer included in the analysis after the date of removal.

**Figure 3 vetsci-10-00421-f003:**
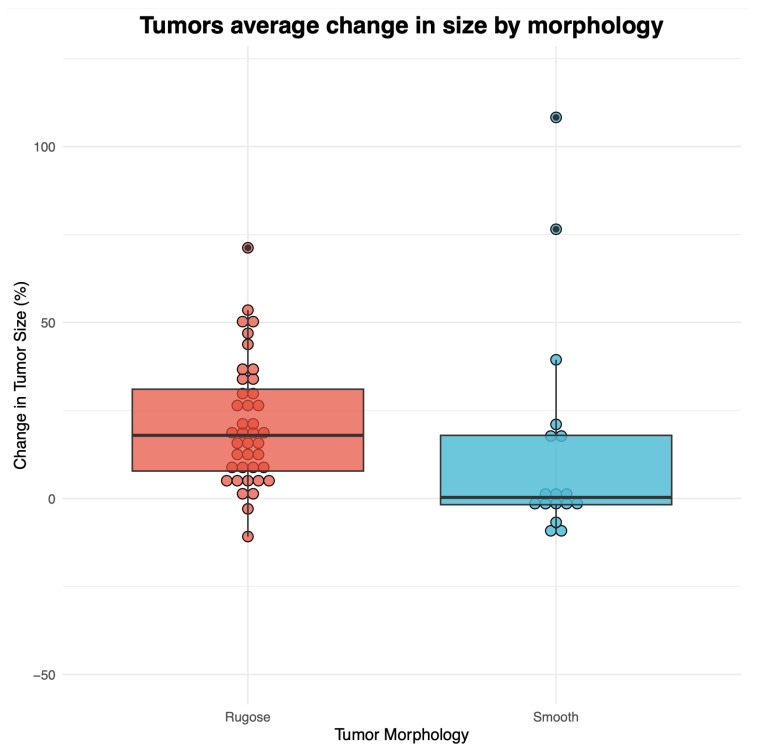
Boxplot of biweekly mean change in size of rugose and smooth FP tumors identified across nine rehabilitating green turtles (*Chelonia mydas*). Regression and progression patterns calculated as biweekly average change in size (shown as %) are shown on the *y*-axis for both smooth and rugose tumors. Circles represent specific data points. Tumor morphology is shown on the *x*-axis. The bold black line inside the boxes represents median tumor change in size for each type of tumor: rugose morphology on the left side (red) and smooth morphology on the right side (blue).

**Table 1 vetsci-10-00421-t001:** Characteristics of nine rehabilitating green turtles (*Chelonia mydas*) and respective rugose and smooth FP tumors. Table showing straight carapace length (cm), weight (kg), number of smooth and rugose FP tumors, as well as total number of FP tumors for each study turtle. The table includes the calculated average biweekly change in tumor size among tumors within one category (i.e., smooth or rugose) and standard deviation. The range of biweekly change in size among tumors within one morphological category is indicated in parenthesis.

Turtle ID	Straight Carapace Length (cm)	Weight (kg)	# Smooth Tumors	Change in Size of Smooth Tumors (%)	# Rugose Tumors	Change in Size of Rugose Tumors (%)	Total # FP Tumors
*Cm* 1	NA	4	1	76.5	2	48.4 ± 2.1 (46.9–50.0)	3
*Cm* 2	64.6	34	1	−6.76	3	5.53 ± 14.5 (−10.8–17.0)	4
*Cm* 3	32.8	4	1	−8.1	14	23.5 ± 20.4 (3.6–71.2)	15
*Cm* 4	29.2	2.8	4	10.7 ± 10 (1.7–21.1)	14	16.1 ± 11.1 (−2.9–43.7)	18
*Cm* 5	35.1	4.4	1	−0.1	1	50.4	2
*Cm* 6	32.5	4.1	1	39.3	3	33.9 ± 2.9 (30.5–36.1)	4
*Cm* 7	38.4	5.0	1	17.5	1	7.3	2
*Cm* 8	42.8	7.5	6	−2.6 ± 3.7 (−10.1–0.3)	1	0.4	7
*Cm* 9	33.6	4.6	1	108.2	1	13.9	2

**Table 2 vetsci-10-00421-t002:** Anatomical distribution of rugose and smooth-textured FP-associated tumors evaluated in nine green turtles (*Chelonia mydas*) under rehabilitative care in St. Augustine, Florida. Tumor locations were divided into four main anatomical regions: along flipper (front or hind flippers), base of flipper (front or hind flippers), ventral neck, and plastron.

Tumor Anatomic Region	Rugose	Smooth
Along flipper	4	1
Base of flipper	26	14
Ventral neck	1	0
Plastron	10	1

## Data Availability

Data are contained within the article.

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
