# Peer review of "Green Turtle Fibropapillomatosis: Tumor Morphology and Growth Rate in a Rehabilitation Setting"

_vetsci, 2023, doi:10.3390/vetsci10070421_

Round 1
Reviewer 1 Report
The authors describe a method of calculating tumor growth. The low number of subjects makes the statistical analysis used and the interpretation of the data correct. In my opinion, the study represents an excellent starting point for the clinical study of the pathology, although the difficulty of collecting data in these animals is real.
Unfortunately, Fig. 2 is not complete: it is possible see only empty graphs. Also it seems that the values of the standard deviations are missing (eg linea 148)
I just have a few curiosity:
Have the surgically removed tumors had recurrences?
was autopsy performed on Cm3 and Cm4 which had multiple FP tumors?
If yes, were lesions different from those reported for Cm5 and Cm6?
Author Response
Dear Reviewer #1,
We would like to thank you for the insight and help given to us through this review. Below is a question-by-question response document, following the format reported below.
Thank you for your valuable contribution.
Kindest regards,
The authors
Reviewer 2 Report
Internal tumors also occur sometimes, including on heart, lungs, and kidneys. (there are other organs that FP has been observed so perhaps say for example or such as or list all sites where tumors have been documented.
No red or blue lines are showing up in Figure 2 on the version I am reviewing
Line 210 smooth (N=17) tumors across the examined turtles[11] . Of all tumors analyzed in our study (delete space between turtles and period
Red and blue not showing up in Figure 3
The manuscript is well written and is worthy of publication. There are very few edits needed.
Author Response
Dear Reviewer #2,
We would like to thank you for the insight and help given to us through this review. Below is a question-by-question response document, following the format reported below.
Thank you for your valuable contribution.
Kindest regards,
The authors
Reviewer 3 Report
There is only one comment about the manuscript, I don't know if it was something wrong with how I downloaded the manuscript to read but I wasn't able to see Figures 2 and 3, there were no lines or colors as explained in the title of each figure.
I suggest checking that and maybe trying to modify the settings of these graphs so readers can download the manuscript without this problem. There are two comments on this inside the pdf.

Author Response
Dear Reviewer #3,
We would like to thank you for the insight and help given to us through this review. Below is a question-by-question response document, following the format reported below.
Thank you for your valuable contribution.
Kindest regards,
The authors